# RETRACTED: Ameliorative Properties of Boronic Compounds in In Vitro and In Vivo Models of Alzheimer’s Disease

**DOI:** 10.3390/ijms21186664

**Published:** 2020-09-11

**Authors:** Panchanan Maiti, Jayeeta Manna, Zoe N. Burch, Denise B. Flaherty, Joseph D. Larkin, Gary L. Dunbar

**Affiliations:** 1Field Neurosciences Institute Laboratory for Restorative Neurology, Central Michigan University, Mt. Pleasant, MI 48859, USA; 2Department of Psychology, Central Michigan University, Mt. Pleasant, MI 48859, USA; 3Program in Neuroscience, Central Michigan University, Mt. Pleasant, MI 48859, USA; 4Field Neurosciences Institute, Ascension St. Mary, Saginaw, MI 48604, USA; jayeeta.manna@ascension.org; 5College of Health and Human Services, Saginaw Valley State University, Saginaw, MI 48604, USA; 6Department of Biology, Eckerd College, St. Petersburg, FL 33711, USA; znburch@eckerd.edu (Z.N.B.); flaherdb@eckerd.edu (D.B.F.); 7Department of Chemistry, Eckerd College, St. Petersburg, FL 33711, USA; larkinjd@eckerd.edu

**Keywords:** Alzheimer’s disease, amyloid beta protein, neurodegeneration, boron compound, neuroinflammation, amyloid plaque, *C. elegans*

## Abstract

Alzheimer’s disease (AD) is characterized by amyloid (Aβ) aggregation, hyperphosphorylated tau, neuroinflammation, and severe memory deficits. Reports that certain boronic compounds can reduce amyloid accumulation and neuroinflammation prompted us to compare trans-2-phenyl-vinyl-boronic-acid-MIDA-ester (TPVA) and trans-beta-styryl-boronic-acid (TBSA) as treatments of deficits in in vitro and in vivo models of AD. We hypothesized that these compounds would reduce neuropathological deficits in cell-culture and animal models of AD. Using a dot-blot assay and cultured N_2_a cells, we observed that TBSA inhibited Aβ42 aggregation and increased cell survival more effectively than did TPVA. These TBSA-induced benefits were extended to *C. elegans* expressing Aβ42 and to the 5xFAD mouse model of AD. Oral administration of 0.5 mg/kg dose of TBSA or an equivalent amount of methylcellulose vehicle to groups of six- and 12-month-old 5xFAD or wild-type mice over a two-month period prevented recognition- and spatial-memory deficits in the novel-object recognition and Morris-water-maze memory tasks, respectively, and reduced the number of pyknotic and degenerated cells, Aβ plaques, and GFAP and Iba-1 immunoreactivity in the hippocampus and cortex of these mice. These findings indicate that TBSA exerts neuroprotective properties by decreasing amyloid plaque burden and neuroinflammation, thereby preventing neuronal death and preserving memory function in the 5xFAD mice.

## 1. Introduction

Neurodegeneration, synaptic loss, and increased neuroinflammation are the common pathological symptoms observed in Alzheimer’s disease (AD). Accumulation of misfolded amyloid beta protein (Aβ) might be the principal cause and is strongly associated with these events. Several experimental, clinical, biochemical, and molecular studies have correlated these pathological symptoms with memory impairment in AD [1,2]. Therefore, decreasing neuroinflammation or disaggregating the misfolded Aβ are proposed strategies for preventing neurodegeneration and preserving cognitive function in AD. Because of the complexity of the pathological profile observed in AD, it may not be feasible to reduce major AD symptoms using a single drug. Although the US Food and Drug Administration (FDA) has approved several drugs for treating AD, most of them provide only modest improvements in memory and cognitive function and are unable to prevent progressive neuronal death. Therefore, developing new drugs for the treatment of AD is a challenging task for researchers. One area of drug development receiving renewed attention is the use of boron compounds which, because of their anti-inflammatory properties, are being proposed as potential treatments for reducing AD-induced neuroinflammation and cognitive deficits [3,4,5,6].

Boron is a solid, nonmetallic element (atomic number 5) in group 13 of the periodic table. It is one of the most important minerals found in food and in the environment. It has pleiotropic effects, including building strong bones, muscles, treating osteoarthritis, and increasing testosterone levels in humans [7]. Because of its role in improving cognitive skills and muscle coordination, some people take it as dietary supplement [6], which has been an impetus for using it in the development of pharmaceutical drugs [7]. Several boron derivatives have been tested for different diseases [3]. For example, bortezomib, a proteasome inhibitor, has been used to prevent several malignancies [8]. Cyclic boron derivatives, such as tavaborole, have been developed as antifungal agents and are FDA-approved topical treatments for onychomycosis [9,10]. Similarly, benzoxaborole SCYX-7158 has been tested clinically for the treatment of stage 2 human African trypanosomiasis [11]. Furthermore, a group of boron compounds have shown to be strong inhibitors of phosphodiesterase 4 enzyme (PDE4) and inflammation-related cytokine release [12].

Although studies have demonstrated that boron compounds have strong immunomodulatory functions [13,14], only, a few studies have investigated their potential to treat neurodegenerative diseases, such as AD. Recently, Kucukdogru and colleagues reported that boron nitride nanoparticles have neuroprotective effects in the experimental Parkinson’s disease model against MPP+-induced apoptosis [3]. Similarly, Lu and colleagues discovered a series of boron-containing compounds to be Aβ aggregation inhibitors and antioxidants for the treatment of AD [4]. In our previous study, we found evidence of the reduction of Aβ plaque-induced neurodegeneration in *C. elegans* overexpressed with Aβ42 after treatment with trans-2-phenylvinylboronic acid (TPVA; unpublished data). The TPVA MIDA ester and TBSA are monomeric, free-flowing, crystalline solids which are stable at room temperature. They are mainly used for palladium (Pd)-catalyzed Suzuki-Miyaura coupling reactions, diastereoselective synthesis, and rhodium (Rh)-catalyzed intramolecular amination of aryl azides. They are also used as reagents in preparation of optically active unsaturated amino acids by diastereoselectivity and isomerization. Our preliminary observation in *C. elegans* Aβ42-expressing models with boronic compounds show promising results which prompted us to investigate the mechanistic details of these two boronic compounds.

Because of the limited number of neurons (~302 neurons), making it easy to track and manipulate their function, *Caenorhabditis elegans* have been used by many researchers to test neuronal vulnerability. To investigate the effect of different anti-amyloid compounds (such as boron) on Aβ toxicity in neurons, researchers used a transgenic *C. elegans* model that expressed Aβ 1–42 in glutamatergic neurons, which are particularly vulnerable in AD [15]. In the present study, we utilized a *C. elegans* Aβ42-expressing model to test our hypotheses.

In the present study, we investigated the effects of trans 2-phenyl vinyl-boronic acid MIDA ester (TPVA) and trans-beta-styryl-boronic acid (TBSA) on Aβ aggregation and neurotoxicity in vitro. In addition, survival and morphological changes in a *C. elegans* model of AD, as well as Aβ plaque loads, neuroinflammatory markers, and behavioral outcomes in 5xFAD mice were explored after treatment with TBSA. We observed that boronic compounds significantly inhibited Aβ aggregation and neurotoxicity in vitro. Increased survival and neuroprotection was observed in the *C. elegans* model of AD, while decreases in amyloid plaque, neuronal death, and neuroinflammation, along with a partial reduction in memory deficits were also observed in the 5xFAD mouse model of AD after two months of treatment with TBSA.

## 2. Results

### 2.1. Trans-Beta-Styryl-Boronic Acid (TBSA), But Not Trans 2-Phenyl Vinylboronic Acid (TPVA) MIDA Ester Inhibited Aβ42 Aggregation and Toxicity In Vitro

Aβ42 aggregation inhibition was monitored using 6E10 antibodies after treatment with TBSA and TPVA for 24 h with different concentrations. We used 5, 10- and 20 for TPVA (in µg/mL, B) and 5-, 10, 20-, 50 and 100 for TBSA (in µg/mL, D) for our dot blot assay, separately. Then, in another experiment, we used 1-, 5- and 10 µg of both TBSA and TPVA ((in µg/mL, F). Both TPVA and TBSA inhibited Aβ42 aggregation with 10- and 20- µg concentrations. We observed that TBSA inhibited Aβ42 aggregation significantly at 1 mg and 0.5 mg concentrations, but not with the 0.1 mg dose, whereas TPVA increased Aβ42 aggregation, as shown by probing with 6E10 (Figure 1E,F). Similarly, TBSA (0.5 mg) significantly protected against Aβ42-induced neuronal cell death, but TPVA did not (Figure 1G,H).

### 2.2. Trans-Beta-Styryl-Boronic Acid (TBSA) Treatment Was Tolerated In Vivo and Ameliorated Neurodegeneration in “AD” Aβ-Expressing C. elegans

After 17 days, treatment with TBSA did not decrease the viability or survival of the AD (UA198) or control (Peat) animals (Figure 2A,B). At day 17, survival of AD animals was improved in TBSA-treated populations compared to untreated animals (*p* < 0.05, single-tailed t-test) (Figure 2A,B). Furthermore, the number of living tail neurons in the treated AD animals was significantly greater than in the untreated AD animals (one-tailed t-test, *p* < 0.01) (Figure 2C,D). Healthy adult animals normally have 5 GFP-expressing neurons in the tail region in these strains.

### 2.3. Body Weights of Mice

There was no significant change in body weight, and no death of animals for either the 6- or 12-month-old groups of mice during the course of treatments (data not shown).

### 2.4. Trans-Beta-Styryl-Boronic Acid (TBSA) Treatment Moderately Decreased Aβ Plaque Load in 5xFAD Mice

After two months of TBSA treatments, the brain sections of the 5xFAD mice were stained for Aβ plaques with curcumin (1 µM), as demonstrated previously [16,17]. The number of Aβ plaques was quantified in the cortex, CA1, CA3, and DG areas of hippocampus in both 6- and 12-month-old 5xFAD mice receiving TBSA or vehicle. The number of Aβ plaques was significantly higher the cortex, CA1, CA3, and DG of hippocampus in vehicle-treated 5xFAD mice, while treatments of TBSA reduced this increase in plaque numbers in the 6- and 12-month-old groups of all areas sampled, except in the cortex and DG area of the 12-month-old group of mice (Figure 3A–E).

### 2.5. Trans-Beta-Styryl-Boronic Acid (TBSA) Treatment Reduced Pyknotic Cells in Different Brain Areas of 5xFAD Mice

One of the aims of this study was to investigate whether a chronic 2-month treatment of TBSA protects the neuronal morphology in cortical and hippocampal subfields. Paraffin-embedded tissue sections were stained with 0.1% cresyl violet and pyknotic or tangle-like neurons were counted within the pyramidal cell layers of the cortex, the CA1, and CA3 subfields of the hippocampus. In the case of cortex and the CA1 and CA3 areas of the hippocampus, a significant increase in the percentage of pyknotic or tangle-like cells was observed in 5xFAD mice, whereas treatment with TBSA partially reduced the percentage of pyknotic cells in both 6- and 12-month-old mice when compared to 5xFAD + vehicle-treated mice (Figure 4B–G). Damaged cells were more prevalent in the case of 12-month-old mice, relative to the 6-month-old group in all three brain regions.

### 2.6. Trans-Beta-Styryl-Boronic Acid (TBSA) Treatment Reduced Astrocyte Activation in Different Brain Areas of 5xFAD Mice

We were interested in determining whether a chronic 2-month treatment of TBSA could reduce neuroinflammation in 5xFAD mice in cortical and hippocampal subfields. Paraffin-embedded tissue sections were immunolabeled with astrocyte (GFAP) and microglial (Iba-1) markers. The GFAP-IR and Iba-l-IR cells were counted in the cortex, the CA1, and CA3 subfields of the hippocampus. We observed a significant increase in the number of GFAP-IR in all these areas in 5xFAD mice, whereas treatment with TBSA significantly decreased the number of GFAP-IR in 12-month-old mice in all areas studied (Figure 5A–G).

### 2.7. Trans-Beta-Styryl-Boronic Acid (TBSA) Treatment Reduced Microglial Activation in Different Brain Areas of 5xFAD Mice

Paraffin-embedded tissue sections were immunolabeled with microglial marker (Iba-1) in the cortex and hippocampus of 5xFAD mice after treatment with TBSA to investigate whether TBSA has any inhibitory role. Our morphometric data revealed more Iba-1-IR-positive cells in 5xFAD mice in all sampled areas in comparison to WT + vehicle treated mice, whereas treatment with TBSA significantly reduced the number of Iba-1-IR cells in both the 6- and 12-month-old groups (Figure 6A–G). A greater reduction was observed in the case 12-month-old group in comparison to 6-month-old mice in all these areas sampled (Figure 6C,E,G).

### 2.8. Effect of Trans-Beta-Styryl-Boronic Acid (TBSA) Treatment on Open Field Test in 5xFAD Mice

To assess spontaneous locomotor activity and anxiety levels in 5xFAD mice, an open-field test was performed before and after treatment with TBSA. As a measure of anxiety, we counted the number of fecal boli from each group of mice and found a significant increase in the 5xFAD mice in comparison to WT in pretreated mice at both 6- and 12-months of age, with treatments of TBSA showing reduced fecal boli accounts in the 6-month-old 5xFAD mice (Figure 7A,B). However, we did not observe any significant between-group differences in distance travelled (Figure 7C,D) or locomotor average speed (Figure 7E,F) in the open field for either group.

### 2.9. Effect of Trans-Beta-Styryl-Boronic Acid (TBSA) Treatment on Novel Objective Recognition Tasks in 5xFAD Mice

The novel object recognition (NOR) test was used to investigate the recognition memories of mice for novel objects. We observed that 12-month-old, but not 6-month-old, 5xFAD mice spent significantly less time exploring the novel object in comparison to WT mice, whereas treatment with TBSA had no effect on this measure of recognition memory deficits (Figure 8A,B). However, TBSA treatments exerted some preservation of recognition memory in the 12-month-old TBSA-treated 5xFAD mice as expressed by the exploration index (Figure 8C) and discrimination index (Figure 8D).

### 2.10. Effect of Trans-Beta-Styryl-Boronic Acid (TBSA) Treatment on Morris Water Maze (MWM) Performance in 5xFAD Mice

The MWM task was used to investigate whether TBSA treatment could prevent spatial memory loss in 5xFAD mice. Analysis of learning trend over 5 days of training in MWM showed that 6- and 12-month-old-vehicle-treated 5xFAD mice took significantly longer to reach the platform on days 4 and 5 (Figure 9A,B), in comparison to WT + Vehicle treated mice, that this was prevented by TBSA treatments (Figure 9A,B). Similarly, the 12-month-old, but not the 6-month-old, vehicle-treated 5xFAD mice swam significantly more distance to reach the platform in comparison to WT mice (Figure 9C,D); this deficit was also prevented by the TBSA treatment (Figure 9D). No significant changes in overall swim speed (cm/s) were observed in 12-month-old groups of mice; however, 6-month-old WT + TBSA-treated mice showed relatively faster swim speeds in comparison to all other groups (Figure 9E).

### 2.11. Effect of Trans-Beta-Styryl-Boronic Acid (TBSA) Treatment on Spatial Memory in 5xFAD Mice

The mean number of entries to the target quadrant was significantly higher in the 6-month-old 5xFAD + TBSA and WT + TBSA groups in comparison to 5xFAD and WT + Vehicle-treated mice (Figure 10B,C). In contrast, the 12-month group, WT + TBSA-treated mice had significantly more entries into the target quadrant in comparison to WT + Vehicle, 5xFAD + Vehicle and 5xFAD + TBSA-treated mice (Figure 10C). The 6-month-old vehicle-treated 5xFAD mice spent less time in the target quadrant in comparison to 5xFAD + TBSA and WT + Vehicle and WT + TBSA (Figure 10B,D), whereas no group differences were observed in the case of 12-month-old mice (Figure 10D). Latency of first entry to the target quadrant which previously contained the platform (Figure 10A) was significantly higher for both the 6- and 12-month-old, vehicle-treated 5xFAD mice in comparison with their age-matched WT + Vehicle-treated and 5xFAD mice treated with TBSA (Figure 10E). No significant between-group differences for distance from the target quadrant was observed in the case of 6-month group of mice, but vehicle-treated, but not TBSA-treated, 5xFAD mice had mean distances which were farther from the former location of the platform in comparison to WT mice (Figure 10F).

## 3. Discussion

Disturbances of protein homeostasis and increased neuroinflammation are the hallmark pathologies observed in the brains of AD patients. These pathological changes are strongly linked with neurodegeneration, synaptic loss, as well as cognitive dysfunction. Therefore, reducing neuroinflammation and decreasing misfolded amyloid protein could be a viable approach to mitigate cognitive dysfunction in AD. In the present study, we investigated the effects of chronic administration of a boronic compound, trans-beta-styryl-boronic acid (TBSA) on: (i) survival and neurodegeneration in *C. elegans* expressing Aβ42; (ii) neuronal morphology in both *C. elegans* expressing Aβ42 and 5xFAD mice; (iii) amyloid plaque load in 5xFAD mice; (iv) neuroinflammatory markers in 5xFAD mice; and (v) behavioral deficits in 5xFAD mice. We observed that in aged *C. elegans* expressing Aβ42 (17 days old), survival was improved with TBSA treatment, as was neuronal health. In 5xFAD mice, a significant decrease in neuroinflammatory markers, amyloid plaque load, and prevention of neuronal damage in the cortex and hippocampus, along with modest protection against cognitive dysfunction in 5xFAD mice was observed after treatment with TBSA.

Several anti-amyloid, anti-inflammatory drugs and small molecules have been tested to inhibit amyloid aggregation and neuroinflammation in AD [18,19]. Some of these molecules prevented Aβ aggregation and neuroinflammation, but were unable to protect against progressive neurodegeneration, as well as loss of cognitive function in AD [20]; therefore, none of these compounds proved successful in clinical AD trials. In the present study, we tested the efficacy of two different boron compounds against Aβ-induced neurodegeneration. Boron is a nonmetallic element found in food and the environment, and it has a high affinity to bind to oxygen [21]. It is naturally electron deficient, thus it is well suited to unique chemical reactions. It readily accepts electrons from nucleophiles like the hydroxyl group in serine, making boronic drugs highly effective serine protease inhibitors [10]. Although as a trace element, boron has yet to be established as an essential nutrient for humans, recent experimental and clinical studies suggest that it might be an important mineral for cell membrane function [7]. Because of their anti-inflammatory properties, boron compounds have been used as dietary supplements for the treatment of neuroinflammation and in neurodegeneration [4]. Boron compounds have been tested to treat a host of disorders, from arthritis to vascular dysfunction [13,22]. For example, Bortezomib acts as a potent ubiquitin proteasome inhibitor and is commonly used to treat myeloma [8]. Similarly, Vaborbactam, a β-lactamase inhibitor has been used to fight bacterial infections, such as treatment for complicated urinary tract infections [23]. Furthermore, sodium perborate has been used in whitening discolored teeth, disinfecting medical instruments, as an ointment and for the treatment of poison ivy dermatitis [24].

However, the use of boron-based compounds has not been tested as a therapy to reduce AD pathology. Penland and colleagues used electrophysiology and cognitive performance tests and reported that dietary boron may play important roles for brain function and cognitive performance [6]. Previous experimental data suggest that boronic acid may have a high affinity for binding directly to amyloid precursor protein (APP) [25]. In addition, due to their high pKa value (~9–10), boron compounds remain unionized at physiological pH, which favor their preferential binding to Aβ; thus, they have been utilized in detection of Aβ plaque aggregation in AD [25,26].

Based on the above findings, we were interested in testing the efficacy of different boron compounds in animal models of AD. Boronic compounds have not been explored much in the field of AD research, especially in experimental AD models. In our previous study, we found some evidence of a reduction of neurodegeneration in a *C. elegans* which genetically overexpressed Aβ42. However, the mechanism in which the boron compounds interact with Aβ remains unknown. In the present study, we investigated whether boron compounds can increase survival and neuronal integrity in a *C. elegans,* which is genetically modified to overexpress Aβ42 and protect against neuroinflammation, amyloid plaque loads, and cognitive dysfunction in the 5xFAD mouse model of AD.

Initially, we compared the Aβ42 aggregation inhibition capability of trans 2-phenyl vinylboronic acid (TPVA) MIDA ester and trans-beta-styryl-boronic acid (TBSA) using a dot blot assay. To determine the relative efficacy as an Aβ42 aggregation inhibitor between these two boronic compounds, we used a synthesized Aβ42 peptide (at 10 µM) and allowed it to aggregate with or without these compounds (in µg/mL: 1-, 5- and 10 of both TBSA and TPVA). Interestingly, we found that, at low concentrations (5–10 µg, but not 1.0 µg), both TPVA and TBSA were able to inhibit Aβ42 aggregation, suggesting that low doses of boronic compounds would be sufficient to inhibit Aβ aggregation [27]. We also observed that TBSA was a more potent Aβ42 aggregation inhibitor than TPVA (Figure 1F). We used 6E10, which binds with monomers and oligomers fibrils (1–16 residues of N-terminal fragments). Although oligomer specific (e.g., A11) or fibril specific antibodies (e.g., OC) are preferable for investigating anti-amyloid properties of boron compounds by dot blot assay, other researchers have also used 6E10 to investigate overall Aβ42 aggregation pattern and to make comparisons with other antibodies, such as A11 or OC to investigate oligomer and fibril formation [21,28,29,30,31]. Moreover, to better understand the anti-amyloid properties of boron compounds, we should investigate the changes of secondary structure (random coil to β-pleated sheet) of Aβ42 peptide by circular dichroism spectroscopy (CD spectroscopy), or study the amyloid aggregation size by dynamic light scattering or transmission electron microscopy to investigate the overall morphological changes from monomer to oligomers, protofibril or fibril formation; however, due to limitations in our research facility, we could not use these techniques. The mechanism for this difference is unknown. It is suspected that, being a smaller molecule than TPVA, TBSA may have greater facility for entering the hydrophobic core of the Aβ molecules to prevent its aggregation. Furthermore, the steric hindrance of TPVA might lessen its reaction with Aβ42 molecules more than that of TBSA, although these hypotheses require further investigation.

If these boronic compounds inhibit Aβ42 aggregation, then they might help reduce the toxic effects of Aβ42 and yield increased neuroprotective effects. To test this hypothesis, we cultured mouse neuronal cell lines (N2a) and treated them with Aβ42 as a toxin (10 µM) for 24 h in the presence or absence of either TPVA or TBSA. Interestingly, we observed that TBSA, not by TPVA, protected neuronal death caused by Aβ42 insult (Figure 1H). The toxicity data paralleled our dot blot assay findings, suggesting TBSA might be a better candidate to inhibit Aβ aggregation and neurotoxicity. However, the mechanistic details of neuroprotection caused by TBSA require further investigation.

The therapeutic efficacy of TBSA was then investigated in vivo using a *C. elegans* expressing Aβ42 in its glutamatergic neurons, which has been demonstrated to undergo Aβ-induced neurodegeneration [15,18]. Since TBSA has not been previously investigated in vivo, it was important to evaluate whether or not chronic treatment with it had any impact on the viability and survival of the whole animal. The data show that the TBSA treatment did not appear to influence animal survival in control animals or AD animals up to 15 days posthatching (Figure 2A). At the old age of 17 days, however, TBSA treated AD mice did have a statistically significant improvement in survival compared to the untreated AD animals (Figure 2A,B). It is possible that this improvement in survival can be explained by the statistically significant improvement in neuronal health seen with treatment with TBSA (Figure 2C,D). As the data show, examination of the number of live GFP-expressing glutamatergic neurons that persist in the tail region of animals of advanced age provides a robust way to assess neuronal health in Aβ-expressing neurons (Figure 2C). Healthy young adult to adult animals normally express 5 GFP-positive neurons in their tails. Beyond the difference in the sheer number of living neurons (Figure 2D), micrographs also reveal that many neurons in the untreated animals, while still present, are more degenerated, as evidenced by their reduced size and the increase in aberrant varicosities (Figure 2C). Again, the mechanism of action for TBSA to help protect the neurons from degeneration needs further investigation. It is true that the survival of the Peat-4::GFP control animals treated with TBSA had the lowest survival rate, i.e., 17 days of age, but that survival was not statistically different from Peat-4::GFP untreated animals, nor the untreated *C. elegans* expressing Aβ42. Hence, only the *C. elegans* expressing Aβ42 treated with TBSA had a stronger survival rate by day 17. We reason that if Aβ were actually beneficial, then we would expect the *C. elegans* expressing Aβ42 untreated animals to either not be statistically different from the *C. elegans* expressing Aβ42-treated and/or to be statistically different from the Peat-4::GFP control worms. We hypothesize that the TBSA drug confers a potent support of health in the Aβ42-expressing animals, and that in the absence of Aβ, the drug may confer a low level of toxicity without this preferred molecular target.

Based on these findings and previous experimental observations [5,6], we decided to use TBSA as a potential therapy in 5xFAD mice. After 2 months of oral gavage (0.5 mg/kg), we found a significant decrease in Aβ plaque burden in several brain areas, such as the cortex, and the CA1, CA3, and DG regions of hippocampus (Figure 3), suggesting that the TBSA crosses the blood brain barrier, permeates into the brain tissue, and inhibits amyloidogenic pathways, either by reducing Aβ production or preventing its aggregation. Although we did not measure the boron levels in the TBSA-treated mice brain tissue, previous studies have shown that boron compounds can cross the blood brain barrier and penetrate the brain tissue [32] and decrease inflammatory responses [33]. Combined with previous research showing efficacy of boron compounds in the brain [3,4,5,6,7], our findings of the efficacy of TBSA in the present study strongly suggest that TBSA is sufficiently permeable to brain tissue to reduce AD pathology.

We also investigated whether TBSA treatments preserved neuronal morphology in affected areas of the 5xFAD brain. We used cresyl violet (CV) stains in paraffin-embedded tissue sections to investigate overall neuronal morphology and number of pyknotic cells. Our morphometric data suggested that TBSA treatment significantly reduced the number of pyknotic, or tangle-like, cells (Figure 4) in the cortex and in the CA1 and CA3 subfields of the hippocampus in the 5xFAD mouse brain. These findings are in concert with the findings of decreased Aβ plaque burden in TBSA-treated 5xFAD mice, suggesting TBSA may have an inhibitory role on Aβ production.

Inflammation is an important pathological correlate with Aβ accumulation. Reducing the spread of inflammatory response requires brain immune cells to engulf or sequester the amyloid plaques. Therefore, the activation of astrocytes and microglia is a primary event in AD pathology. Given that boron compounds have a vast array of anti-inflammatory actions, we investigated whether TBSA could prevent neuroinflammation in 5xFAD mice, especially in the most affected brain areas of this AD mouse model. We measured the number of activated astrocytes (GFAP-IR) and microglia (Iba-1-IR) in affected areas. We observed a clear inhibition of astrocytic and microglial activation, as the number of GFAP-IR and Iba-1-IR was significantly decreased in all the areas studied (Figure 5 and Figure 6). Although we did not categorize which type of microglia were inhibited by TBSA treatment, our findings are in concert with others which demonstrate that boron compounds can attenuate inflammatory responses in different animal models of neurological diseases [14]. How astrocytes and microglia become inhibited by TBSA treatment is unknown, but in vitro studies have revealed that boron can affect the production of inflammatory cytokines by cartilage cells, which are involved in the inflammatory response [34]. In addition, boron, as boric acid, can stimulate the synthesis and release of tumor necrosis factor-α (TNF-α) in chick embryo cartilage and fibroblasts [34], suggesting that it could decrease tau phosphorylation in AD [21].

Very few studies have reported that daily intake of boron compounds can have cognitive effects, especially for improving learning, memory, and attention in normal individuals [6]. In the present study, we attempted to investigate the effects of TBSA on counteracting AD-associated cognitive impairments in 5xFAD mice. Using the NOR task, we measured recognition memory, which is commonly impaired in AD patients. We observed a significant deterioration of both the discrimination index (Figure 8D) and the exploration index (Figure 8C) in 12-month-old 5xFAD mice, which was mitigated by TBSA treatments (Figure 8C,D). This was probably a mnemonic effect and not the result of increased anxiety or activity levels, as measures of fecal boli and movements in the open field did not reveal persistent deficits in the 5xFAD mice.

Impairment of the spatial memory is common in AD patients [35]. To investigate the role of TBSA on spatial memory tasks in 5xFAD mice, we performed the Morris-water-maze (MWM) task. The significant increase in escape latency (on days 4 and 5) and pathlength (on day 5) to find the hidden platform in the 12-month-old vehicle-treated 5xFAD mice was prevented by the TBSA treatments (Figure 9). Because the average swim speed was equivalent between both the vehicle- and TBAS-treated 5xFAD mice (Figure 9E), the differences accurately reflect spatial memory abilities. In addition, during the probe trial, TBSA-treated 5xFAD mice took less time to enter to the target quadrant and kept their mean distance closer to the target quadrant compared to the vehicle-treated 5xFAD mice (Figure 10), suggesting TBSA protects against spatial memory deficits in 5xFAD mice.

The modest TBSA-induced protection against cognitive deficits in the 5xFAD mice may be due to the TBSA-induced reduction in amyloid plaques, reduced neuroinflammation, and/or reduced neurodegeneration (Figure 11). Previous reports have suggested that boronic compounds bind directly to amyloid precursor protein (APP) or Aβ [22,23]; we found a significant reduction in Aβ aggregation in vitro and decreased plaques levels in 5xFAD mice, which could reduce the overall neuroinflammation and neuronal death, although the extent to which any of these and other possible mechanisms of action can explain the therapeutic effects of TBSA requires further investigation, especially in the context of optimal dose and duration of TBSA treatments. To our knowledge, this is the first study using any boron compound to treat deficits in the 5xFAD mouse model of AD. Extending these findings to other models may provide a stronger basis for the use of boron compounds in clinical trials.

Our findings support the work of Penland, who found that a dietary boron intake increases overall brain function and cognitive performances in humans [6,7]. Similarly, Nielsen and Meacham reported that boron supplementation after boron deprivation resulted in improved functioning, including less drowsiness and mental alertness, improved psychomotor skills (e.g., motor speed and dexterity), and improved cognitive processing (e.g., attention and short-term memory) in older men and women as shown by electroencephalograms [6,7]. In another study, Nielsen and Penland reported that boron deprivation in rats reduced the number, distance, and time of horizontal movements, front entries, margin distance, and vertical breaks and jumps in assessments of spontaneous activity compared with performance of rats given boron supplements [5]. Our findings confirm earlier work and extend previous research showing that boron compounds can counteract both impaired recognition and spatial memory dysfunction.

## 4. Materials and Methods

### 4.1. Chemicals

Trans 2-phenyl vinylboronic acid (TPVA) MIDA ester (catalog no: 699292-5G), Aβ42 peptide, curcumin (Cur, ~65% pure), 1,1,1,3,3,3-hexafluoroisopropanol (HFIP), penicillin/streptomycin, fluoro-mount media and other accessory chemicals were procured from Sigma (St. Louis, MO, USA). Trans-beta-styryl-boronic acid (TBSA) was purchased from Alfa Aesar (97% pure, catalog no: H53227, Haverhill, MA). Aβ antibodies, such as 6E10 were purchased from Bio-Legend (San Diego, CA, USA). Mouse neuronal cell line and were purchased from American type culture collection (ATCC, Manassas, VA, USA). DePeX mounting media was purchased from BDH (Radnor, PA, USA). Minimum essential medium (MEM), fetal bovine serum (FBS) were purchased from GIBCO (Grand Island, NY, USA). Ninety-six-well plate was purchased from COSTAR Corning, (Millipore-Sigma, St. Louis, MO, USA). Horseradish peroxidase-conjugated secondary antibody was purchased from Santa Cruz Biotech (Santacruz, CA, USA). The GFAP antibody was procured from Cell Signaling Technology (Danvers, MA, USA) and Iba-1 was purchased from FUJIFILM Wako Chemicals U.S.A. Corporation (Richmond, VA, USA).

### 4.2. Inhibition of Aβ-Aggregation by Boronic Compounds Using Dot-Blot Assays

Dot blot assay was utilized to investigate the inhibitory effects of different boronic compounds, as described previously [16,17]. Briefly, Aβ42 peptide was dissolved in 1,1,1,3,3,3-hexafluoro isopropanol (HFIP), sonicated for 1 min and allowed to solubilize for 30 min at room temperature. Then the HFIP was evaporated under a laminar hood, until it was completely dried to make a thin film, at which point it was placed into a speed vacuum for 10 min and then stored at −20 °C. Each HFIP film was dissolved in 60 mM NaOH (final concentration 6 mM) and diluted with Tris-buffer saline (TBS, 0.1 M, pH 7.4, 0.025% NaN_3_) to get the desired peptide concentration (10 μM). Then 20 μL of peptide solution (10 μM) was incubated in the presence or absence of different concentrations of TBSA and TPVA (in µg/mL: 1-, 0.5- and 0.1) for 24 h at 37 °C with gentle shaking (200 rpm). After 24 h of incubation, about 10 μL of peptide solution was spotted on polyvinylidene fluoride (PVDF) membrane (Bio-Rad, CA, USA) and dried. Then the membrane was blocked with 5% nonfat milk in TBS-Tween-20 (TBS-T) at room temperature for 1 h, washed with TBS-T, and probed with 6E10 rabbit polyclonal antibodies (1:1000) in 5% nonfat milk powder in TBS-T, overnight at 4 °C. On the following morning, the blot was washed thoroughly and then probed with antirabbit-horseradish-peroxidase (HRP) conjugated secondary antibody solution (1:20,000) for 1 h at room temperature. The blot was developed with Amersham ECL Prime Western Blot Detection Reagent (GE-Healthcare Life Sciences, PA, USA) for 2–5 min. The dot blots were scanned using a gel documentation system (Bio-Rad, CA, USA), and the optical density of each dot was measured using Image-J software (http://imagej.nih.gov/ij).

### 4.3. Cell Culture

Mouse neuroblastoma cells (N_2_a) (ATCC, Manassas, VA, USA) were used to investigate the neurotoxicity in this study. Briefly, N2a cells were grown in minimum essential medium (MEM), with both cultures containing 10% heat-inactivated fetal bovine serum (FBS) and penicillin/streptomycin (1 μg/mL). The cultures were maintained at 37 °C in a humidified atmosphere at 5% CO_2_. Prior to the experiment, the cells were grown in 96-well plates with fresh MEM, lacking growth factors, as described previously [36].

### 4.4. Treatment of Different Boronic Compounds

TBSA and TPVA were dissolved in methanol and then diluted with fresh MEM. The final methanol concentration in the treated solution was ≤1% (*v*/*v*). Based on the dot blot assays of Aβ aggregation inhibition, we found that 1 and 5–10 µg of TBSA were more potent than any other concentration. Therefore, 0.5 mg/dL of TBSA and TPVA were assessed in the cell viability assays.

### 4.5. Cell Viability by MTT Assay

To investigate the neuroprotective effects of boronic compounds after Aβ42 exposure, we conducted a cell viability test, using a MTT [3-(4,5-dimethylthiazol-2-yl)-2,5-diphenyltetrazolium bromide] assay [36,37,38]. Briefly, the N2a cells were grown in 96-well plates at a density of 1 × 10^6^ cells/mL. On the next day, the cells were treated with freshly prepared concentrations of Aβ42 (10 μM) for 24 h in presence and absence of TBSA and TPVA (5 µg/mL). Following 24 h of treatments, 15 μL of MTT (12 mM) was added to each well and incubated for 4 h at 37 °C. Then, 100 µL of the stop solution (20% SDS in 80% ethyl alcohol) was added and kept for overnight at room temperature. The optical density was measured at 570 nm using a Synergy plate reader (Bio-TEK instruments, Winooski, VT, USA). The results of three independent experiments (8 wells per condition) were normalized to the medium control group and expressed as mean ± SEM.

### 4.6. Animals

The *C. elegans* expressing Aβ42 strains, UA198 (baln34[Peat-4: Aβ, *Pmyo*-2:mCherry]; adIs1240[*Peat-4*: GFP) and its corresponding control strain “control” adIs1240[*Peat-4*: GFP were generously provided by the Caldwell lab at the University of Alabama. The AD animals constitutively produced Aβ42 in their glutamatergic neurons under the *eat-4* promoter, a gene which codes for the vesicle filling glutamate transporter. A background transgene of *Peat-4*: GFP was present which provided for direct visualization of the glutamatergic neurons by fluorescence microscopy. Thus, there are normally five GFP-expressing neurons in the tail region of these animals, providing a robust way to determine the extent of neurodegeneration. *C. elegans* strains were cultured in petri dishes containing nematode growth medium (NGM-3 g/L NaCl, 19 g/L bacto-agar, 2.5 g/L bacto-peptone, 5 µg/mL cholesterol, 1 mM CaCl_2_, 1 mM MgSO_4_, 25 mM potassium phosphate at pH 6.0-created from a three-fold mixture of KH_2_PO_4_ to K_2_HPO_4_) agar, with Escherichia coli (strain OP50) grown in the center of the plate as a food source.

For the experiments using mice, six-month-old and one-year-old B6SJL-Tg (APPSwFlLon, PSEN1*M146L*L286V, 1136799Vas/J; Jackson Laboratory, stock no: 34840-JAX/5xFAD) and age-matched wild-type, male and female mice were used. The detailed information of 5xFAD mice are described elsewhere [16,17,38]. All mice were housed at 22°C in the Saginaw Valley State University neuroscience vivarium under a 12-h light/12-h dark, reverse-light cycle with access to food and water *ad libitum*. Transgenic characteristics of all 5xFAD mice were confirmed by genotyping at 3 weeks of age using polymerase chain reaction (PCR), as reported previously [39]. This study was carried out in strict accordance with the protocols approved by the Institutional Animal Care and Use Committee at Saginaw Valley State University (IACUC no-1513829-1). All surgeries were performed under sodium pentobarbital anesthesia (1 mL/4.54 kg body weight), and all efforts were made to minimize animal discomfort.

### 4.7. Treatment of Trans-Beta-Styryl-Boronic Acid (TBSA) Compound

#### 4.7.1. Treatment of TBSA in C. elegans

To assess the toxicity or potential benefits of TBSA in vivo, a total of 244 *C. elegans* expressing Aβ42 were used. Among them, 122 were treated with either 10 µg/mL TBSA or S-media (vehicle). S-Media was prepared in 1L batches by combining 5.9 g NaCl, 50 mL 1M K_2_PO_4_, 1 mL cholesterol (5 mg/mL in ethanol), 10 mL 1M potassium citrate, 10 mL trace metal solution, 3 mL 1M CaCl_2_, and 3 mL 1M MgSO_4_ into 872 mL deionized water. To prepare 10 µg/mL TBSA, 2–5 mg was dissolved in 10 µL of DMSO, then brought to a concentration of 1 mg/mL using S-media. This solution was vortexed then diluted to the 10 µg/mL working concentration. TBSA was applied to the 30 mm plates by pipetting 75 µL onto the bacteria and tilting the plate to cover the surface of the agar. S-media vehicles were prepared and dosed in the same manner. To prepare 10 µg/mL TBSA, 2–5 mg of drug was weighed on analytical balance into the tube used to create the starting stock of the drug that day. The drug was then fully solubilized in 10 µL of DMSO and brought to a concentration of 1 mg/mL within that tube using S-media. This solution was vortexed until all drugs were in solution, and then used to make the 10 µg/mL working concentration. Fresh powdered drug from the desiccated vial was prepared this way every two days, just before the drug was used. TBSA was applied to the 30 mm plates by pipetting 75 µL onto the bacteria and tilting the plate to cover the surface of the agar. S-media with the starting amount of DMSO vehicle was prepared and dosed in the same manner. In this way, animals were never exposed to greater than 0.005% DMSO vehicle, which is 100 to 200-fold below concentrations that have been documented to change the animal’s reproduction or longevity [39,40,41]. Drug and vehicle levels had no impact on OP50 *E. coli* growth or consumption.

Before applying animals to the plates for treatment, they were strictly age-synchronized by allowing a population of gravid adults to lay embryos for 2 h before removing all adults. The synchronized siblings were grown on NGM plates with OP50 *E. coli* for 3 days until they reached the young adult stage. Animals were then transferred by platinum wire pick to treated plates, with no more than 15 animals per plate to avoid crowding. Animals were then assessed for viability every two days and transferred to a freshly dosed plate with food. The experiments were continued through day 17 posthatching to assess aged animals.

#### 4.7.2. Treatment of TBSA in 5xFAD Mice

A total of 48 5xFAD and age-matched wild-type (WT) mice at 6- or 12-months of age (N = 6/group) were administered TBSA (0.5 mg/kg body weight), or equivalent volumes of vehicle (0.5% methylcellulose), via oral gavage, every other day for two months. The dose selection was based on our dot blot and cell viability assays, as well as commonly used doses which were previously reported in the literature [5,6,7]. The mice were randomly divided into eight groups shown in Table 1. The TBSA was dissolved in 0.5% methylcellulose in PBS (0.1 M, pH 7.4). Treatments were initiated 4 days after baseline behavioral tests were completed and continued for 60 days. The same volume of vehicle (0.5% methylcellulose, dissolved in 0.1 M PBS, at pH 7.4) was administered to the vehicle groups as summarized in (Figure 12).

## 5. Behavioral Assays

### 5.1. Open Field Test

The exploratory behavior and spontaneous locomotor activity were measured by open field test (OFT), as described previously [23]. Pre- and post- treatment OFT data were collected on “day 0” and “day 66”, respectively (Figure 12). Details of the OFT protocol were described previously [42]. Briefly, The OFT apparatus consisted of a Plexiglas box (41 cm × 41 cm × 30 cm high) with grids of infrared beams spaced 2.5 cm from the OFT floor. Each of the infrared grids consisted of 16 photobeams in each direction (16 × 16) in which the location of the mouse could be tracked each time the beams in the area were interrupted by movements of the mouse. The automated software connected to the system was used to measure the movement of the mice, as indicated by the number of breaks in the gridded infrared beam system. For OFT, each mouse was placed into the chamber and allowed to explore for 30 min. Total resting time, total distance traveled, and velocity of movement were measured throughout the entire session. In addition, counts of fecal boli were taken as an indirect measure of anxiety.

### 5.2. Novel Object Recognition

The novel object recognition (NOR) test was used to test recognition memory in mice. The detailed protocol was adopted from the studies of Leuptow and colleagues [43]. This task was performed in a grey polyvinyl plastic testing box (40- × 40- × 40- cm). The test consists of two phases: habituation and acquisition. In habituation, the mice were familiarized with the NOR environment for 10 min. On the following day, two circular, white, odorless polypropylene objects (3 cm × 2 cm), which served as familiar objects (FOs), were placed near the center of the box at 14.75 cm from the walls and 25 cm apart from each other for the mice to explore. After 10 min of exploration with the FOs, the mouse was returned to its home cage for 5 min. Then, one of the FOs was replaced with a new object, which was odorless, the same size and color as the FO, but rectangular rather than circular, and was used as the novel object (NO) in this task. The mice were then allowed to explore these objects for another 10 min. The boxes and the objects were cleaned between each trial with 70% ethanol. The entire experiment was video recorded using an overhead camera, which fed the images into a computer with Any Maze software (Columbus, OH). The exploration time (time spent within 5 cm of the object) for the novel object (TN) and familiar object (TF) was measured as recorded by the Any Maze automated software. The exploratory index was measured by subtracting the time (s) spent near the NO from the time spent near the FO using the following equation: (TN-TF). The discrimination index (DI) was calculated with the following equation: (TN − TF)/(TN + TF) × 100. The NOR was conducted on days 2–4 prior to treatment and on days 67–69 after the start of treatment (see Figure 12).

### 5.3. Morris Water Maze (MWM)

The Morris water maze (MWM) task was used to assess spatial memory in mice. The detailed protocols have been described previously [44,45,46]. In this task, mice needed to learn the location of a hidden platform, which was submerged 1.5 cm below the surface of water made opaque by the addition of nontoxic white paint. The MWM tank was a circular pool that was 180 cm in diameter and 153 cm in height and filled with water to a depth of 90 cm and kept at 20–25 °C. The MWM tank was divided into four quadrants: Southeast (SE), Northeast (NE), Northwest (NW) and Southwest (SW), and the platform was kept in the center of the SE quadrant (target quadrant). The foreheads of the mice were marked using black indelible ink from a Permamarker to facilitate tracking of their swim paths. An overhead camera and computer-assisted tracking system (Any-Maze, Wood Dale, IL, USA) recorded the movement of the mouse in the maze, which enabled measurements to be made of the latency (time taken to reach the hidden platform) and path length (distance swam by mice) to find a rectangular transparent platform (10 cm × 10 cm) placed 1.5 cm below the surface of the water in the Southeast (SE) quadrant of the tank. This setup was kept in a 3.6 × 3.3 m room, with four overhead 200-watt mercury lamps for appropriate illumination. All trials in each experiment were performed between 900 and 1200 h. All mice were given four trials per day with a 10-min intertrial interval, and training continued for 5 consecutive days (20 sessions). A trial consisted of gently placing the mouse by hand into the water, facing the wall of the pool at one of four equally spaced starting points (N, S, E and W), and allowing the mouse to swim for 60 s. On the day prior to the first day of testing, the mice were given four habituation trials; if they did not find the hidden platform within the MWM within 60 s, they were guided by hand and allowed to rest on it for 30 s. This procedure was followed during testing, except a different starting point was used on each of the four trials with the order determined randomly. After finding or being guided to the platform, the mouse rested on it for 30 s, after which it was removed and gently towel-dried, before being placed back into its home cage. The dependent measures for this task included the latency and pathlength to find the platform. The average speed of the animal (distance/time) was also calculated. The MWM experiment was conducted on days 70–75 following the start of treatment (Figure 12).

#### Probe Trial

The probe trial for the MWM task was conducted after 5 days of testing. The platform was removed, and the mice were placed in the MWM tank facing the “N” starting point and allowed to swim for 60 s (see Figure 10A). The number of entries, total time, and distance swum by every mouse in each quadrant was recorded by a camera attached above the MWM, which was connected to a computer and analyzed using Any-Maze software (Columbus instruments, Columbus, OH, USA).

### 5.4. Tissue Processing

The number of mice used for studying different parameters is documented in Table 1. For histology and immunofluorescence studies, the mice were deeply anesthetized with an overdose of Fatal-Plus (0.22 mL/kg of body weight, i.p.) and transcardially perfused with 0.1 M cold PBS at pH 7.4, followed by a 4% paraformaldehyde (diluted in 0.1 M PBS at pH 7.4) fixation solution. The brains were extracted from the skull and postfixed in 4% paraformaldehyde and stored at 4 °C until they were processed for immunohistochemistry.

#### 5.4.1. Amyloid β-Plaques Count

To investigate the effect of TBSA on Aβ plaque burden, coronal (40-µm) sections were obtained from the brains of 5xFAD + Vehicle and 5xFAD + TBSA groups of mice using a cryostat. The protocol for Aβ plaque labeling in 5xFAD+vehicle brain tissue was performed using curcumin, as described previously [16,17,24]. Briefly, the sections were washed with freshly prepared PBS, three-times, 2 min each. Then the sections were immersed in 70% ethanol for 2 min at room temperature and stained with curcumin (1 µM, dissolved in methanol and diluted with 70% ethanol) for 10 min at room temperature, on a shaker at 150 rpm. The Cur-derivative solutions were discarded, washed with 70% ethanol, thrice, 2 min each and sections were placed on poly-L-lysine coated glass slides and mounted on a coverslip using the organic mounting media, distyrene plasticizer xylene (DPX). Sections were viewed under a fluorescence microscope (Leica, Wetzlar, Germany), using 480/550 nm excitation/emission filters and images were taken by a 20× objective (total magnification = 200×). Bright green fluorescent dots were considered as amyloid plaques. Curcumin was used to label Aβ plaques, because it labels plaques as efficiently as Aβ-specific antibodies [16,17,24]. To show the specificity of curcumin with Aβ plaques, a 5xFAD brain section was immunolabeled with Aβ-specific antibody (6E10), followed by curcumin solution and checked for colocalization, as described, previously [16,17,24]. The number of amyloid plaques was counted manually in the cortical area (layer IV–V), CA1 and CA3 area of the hippocampus and expressed as the number of Aβ plaques per field. The Aβ plaques were counted only when clearly visible as large fluorescent signals. A minimum of 10 sections, with 20–30 different fields were counted for Aβ plaques and the mean from each group (*n* = 3/group) was calculated from the counts by three researchers, who were blinded to the group identity of the specimens sampled.

#### 5.4.2. Neuronal Morphology by Cresyl Violet Staining

One of the aims of the present study was to investigate whether the boron compound (TBSA) prevented abnormal neuronal morphology in the 5xFAD mice, especially in cortex and hippocampal subfields. To investigate morphological changes in these brain areas, the brains were processed for paraffin embedding and coronal sections were stained with 0.1% cresyl violet as described previously [17,25,26,27]. Briefly, the brains from all groups were dehydrated with graded alcohol and processed with paraffin embedding. Coronal sections (5 µm) were made at +3.70 mm from bregma for cortical section and −3.60 mm from bregma for hippocampal sections using a rotary microtome. Then the sections were stained with 0.1% cresyl violet, as described previously [16,20,22,23]. The sections were washed, dehydrated with graded alcohol, cleared, mounted, and cover-slipped using DePex (BDH, Batavia, IL, USA). Images were taken using a compound light microscope (Olympus) with a 100× objective (total magnification of 1000×). Pyknotic or tangle-like cells were counted manually using Image-J software (http://imagej.nih.gov/ij) and expressed as the number of pyknotic cells per microscopic field. A minimum of 5–7 different sections from each brain area, each with 10 different fields, were used from the cortex and the CA, CA3 and DG regions of the hippocampus to determine the number of pyknotic cells in each group.

#### 5.4.3. Neurodegeneration in C elegans

To assess neurodegeneration, animals were mounted on 2% agarose pads on a microscopic glass slide, immobilized with 2.5 mM sodium azide, and imaged using a Zeiss Axio Imager M2 Research Microscope with the Axiocam 506 and Apotome 2 optical sectioning. Images of the tail neurons of the *C. elegans* were taken using a 40× objective (total magnification 400×) and the Apotome^®^ was used to acquire a z-stack of images. The number of visible fluorescent tail neurons was recorded, and data were averaged. Statistical significance in neuron number was determined using a paired single-tailed t-test. Micrographs were created from a maximum intensity projection of at least seven optical slices at one micron each in order to capture the depth of the body’s tail neurons.

#### 5.4.4. Immunohistochemistry of GFAP and Iba-1

The details of the protocol used for immunoreactivity (IHC) of GFAP and Iba-1 are described elsewhere [17]. Briefly, free floating 40-µm thick coronal sections were blocked with 10% normal goat serum (NGS) in Tris-buffered saline with 0.5% Triton-X100 (TBST) and incubated for 1 h in room temperature. Then, the sections were incubated with GFAP (rabbit monoclonal, 1:250) with 10% NGS in TBST, on a shaker, for overnight at 4oC. The next day, the sections were washed with TBST for 15 min, three times, and incubated with antirabbit secondary antibody-conjugated with Alexa fluorophore 595 (1:1000) and incubated for 1 h at room temperature on a shaker in the dark. After three more washings with PBS, the tissue was counterstained with DAPI for 10 min and washed with distilled water. For Iba-1 immunoreactivity, the sections were blocked with 10% NGS for 1 h on a shaker and then incubated with Iba-1 (rabbit polyclonal, 1:4000) at room temperature for 4 h, followed by overnight incubation on a shaker. The next day, the sections were washed with TBST for 15 min, three times, and incubated with antirabbit secondary antibody-conjugated with Alexa fluorophore 488 (1:1000) for 1 h at room temperature on a shaker in the dark. Then the sections were mounted on poly-L-lysine-coated slides, air dried, and cover-slipped with Fluor-mount media. The signal was detected using fluorescent microscopes (Leica, Germany) with appropriate excitation/emission filters, and images were taken using 20× objectives (total magnification = 200×). The numbers of GFAP-IR and Iba-1-IR were counted from the cortex, CA1 and CA3 area of the hippocampus from all four groups of mice per microscopic field. Three researchers independently counted the GFAP-IR and Iba-1-IR cells from 20–25 images from each group (N = 3–6/group), and the average numbers of GFAP-IR and Iba-1-IR were expressed as mean ± SEM/field.

### 5.5. Statistical Analyses

The behavioral and morphometric data were expressed as mean ± SEM. Unless otherwise noted, all data were analyzed using one-way analysis of variance (ANOVA) with Tukey HSD (honestly significant difference) post hoc tests being conducted when appropriate. Statistical analyses were conducted using the online software available at https://astatsa.com/OneWay_Anova_with_TukeyHSD/. For *C. elegans* experiment, one-tail-*t* test was used to analyze the data. A probability value ≤0.05 was considered statistically significant.

## 6. Conclusions

Our findings indicate that trans-beta-styryl-boronic acid (TBSA) more effectively inhibits Aβ aggregation and neurotoxicity in vitro than trans 2-phenyl vinyl-boronic acid (TPVA) MIDA ester. The TBSA partially protected against neuronal damage and it decreased astrocytic and microglial activation and reduced memory deficits in 5xFAD mice. Collectively, the data from this study indicate that trans-beta-styryl-boronic acid compounds can ameliorate Aβ-induced neuronal damage in animal models of Alzheimer’s disease. Further work is needed to understand the mechanisms of action and to optimize the dose and duration of treatment before a full assessment of its potential for treating AD can be made.

## Figures and Tables

**Figure 1 ijms-21-06664-f001:** Effects of boronic compounds on Aβ42 aggregation and neurotoxicity in vitro. (**A**): structural comparison of two different boron compounds, TPVA and TBSA. (**B**–**G**): Synthesized Aβ42 (10 µM) was disaggregated with hexafluoroisopropanol and dissolved in phosphate buffer saline (PBS, 0.1 M, pH 7.4) and allowed to aggregate in presence or absence of different concentrations of trans 2-phenyl vinyl-boronic acid (TPVA) MIDA ester and trans-beta-styryl-boronic acid (TBSA) for 24 h, after which a dot-blot analysis was performed. About 10 µL of peptide solution was spotted on PVDF membrane and probed with Aβ specific antibody (6E10) and signal was developed using chemiluminescent kit. Images were taken using a gel documentation system. Note that both TBSA and TPVA significantly inhibited the Aβ42 aggregation with 10–20 µg/mL concentration, but TBSA showed greater Aβ42 aggregation inhibition. (**F**,**G**): TBSA treatments resulted in more inhibition of aggregates for both 5- and 10- µg/mL concentrations when compared to treatments with TPVA. (**H**,**I**): Mouse neuronal cell line (N2a) was treated with Aβ42 (10 µM) in presence or absence of TBSA and TPVA for 24 h. Note that the TBSA, but not TPVA, protected Aβ42-induced cell death in vitro (**I**). Values are expressed as means ± SEM; scale bar = 100 µm and is applicable to all images in H. * *p* < 0.05 and ** *p* < 0.01 in comparison to the Aβ42-treated group.

**Figure 2 ijms-21-06664-f002:** Treatments of trans-beta-styryl-boronic acid (TBSA) compound improved survivability and reduced neurodegeneration in a *C. elegans* model expressing Aβ42. The *C. elegans* expressing Aβ42 in their glutamatergic neurons (UA198) and age-matched controls (Peat-4::GFP) were treated with 10 µg/mL TBSA or vehicle every two days out to 17 days old. (**A**): Survival was assessed every two days as a mean percent survived and plotted in a survival analysis curve (* *p* < 0.05 UA198 + TBSA in comparison to all other groups). Starting population per group, *n* = 61. (**B**): Survival was improved in aged (17-day-old) TBSA-treated *C. elegans* model expressing Aβ42 (UA198) compared to untreated animals, and compared to treated and untreated Peat-4::GFP control animals (one-tailed t-test, * *p* < 0.05). (**C**): Representative images show two examples of GFP-expressing glutamatergic neurons in the tails of Peat-4::GFP controls, and UA198 *C. elegans* expressing Aβ42, when untreated (Vehicle) or treated (TBSA), at 17 days of age. In addition to increasing cell survival, the TBSA treatment tended to preserve normal neuronal size and morphology. Magnification = 400x; scale bar = 20 µm. (**D**): The mean number of surviving tail neurons in the tails of Peat-4::GFP controls, versus UA198 *C. elegans* expressing Aβ42, when untreated (Vehicle) or treated with TBSA, at 17 days of age. The average tail neuron number in TBSA treated UA198 *C. elegans* expressing Aβ42 was significantly greater than in the untreated animals (one-tailed t-test, ** *p* < 0.01; *n* = 15). Error bars denote standard error of mean (SEM) in all cases.

**Figure 3 ijms-21-06664-f003:** Treatments of trans-beta-styryl-boronic acid (TBSA) compound inhibited Aβ plaque load in brains of 5xFAD mice. Six- and twelve-month-old 5xFAD and age-matched control mice were treated with TBSA (0.5 mg/kg) or vehicle for 2 months and their brains were perfused with 4% paraformaldehyde. Forty-micron coronal sections were stained with Aβ using a curcumin-based solution (1 µM, dissolved 70% methanol, for 10 min) and the images were taken using a fluorescent microscope with a 20x objective (total magnification = 200x). (**A**): Representative images show the amyloid plaques (arrows) in different brain areas in 5xFAD and 5xFAD mice treated with TBSA. (**B**–**E**): The number of Aβ plaques was significantly decreased by the treatment of TBSA in 6-month-old mice in cortex, CA1 and CA3 areas of hippocampus in comparison to 5xFAD-vehicle treated mice. The number of Aβ plaques was significantly decreased by the treatment of TBSA in 12-month-old mice in CA1 and CA3 areas of the hippocampus in comparison to 5xFAD-vehicle treated mice. Values are expressed as mean ± SEM; scale bar = 100 µm and is applicable to all images Scale bar = 100 µm and is applicable to all images. * *p* < 0.05 and in comparison, to 5xFAD + Vehicle-treated mice.

**Figure 4 ijms-21-06664-f004:** Treatments of trans-beta-styryl-boronic acid (TBSA) compound decreased pyknotic, tangle-like cells in the cortex and hippocampus of 5xFAD mice. Six and twelve-month-old 5xFAD and age-matched control mice were treated with TBSA (0.5 mg/kg) or vehicle for 2 months at which time they were euthanized, and their brains were perfused with 4% paraformaldehyde. The brains were then embedded in paraffin and cut on a rotary microtome into 5-µm coronal sections which were stained with 0.1% cresyl violet. Images were taken through a compound light microscope using a 100x objective (total magnification 1000x). There was a significant increase in the percentage of pyknotic cells in the cortex (**A**–**C**), in CA1 (**A**,**D**,**E**) and CA3 (**A**,**F**,**G**) areas of hippocampus of the vehicle-treated 5xFAD mice, but these increases were mitigated by TBSA treatments in the CA1 and CA3 hippocampus regions. Values are expressed as means ± SEM; scale bar = 100 µm and is applicable to all images. Sale bar = 100 µm and is applicable to all images. * *p* < 0.05 and ** *p* < 0.01 in comparison to WT + Vehicle, 5xFAD + TBSA and WT + TBSA.

**Figure 5 ijms-21-06664-f005:** Treatments of trans-beta-styryl-boronic acid (TBSA) compound reduced astrocyte activation in the cortex and hippocampus in 5xFAD mice. Six- and twelve-month 5xFAD mice and age-matched wild-type mice were treated with TBSA for 2 months, sacrificed, and their brain sections were collected and immunolabeled for GFAP. (**A**): Representative images of GFAP-IR in cortex, CA1 and CA3 area of hippocampus. B–G: Labeled GFAP-IR cells were significantly reduced in the cortex (**B**,**C**), CA1 area (**D**,**E**) and CA3 area (**F**,**G**) of the hippocampus in comparison to vehicle-treated 5xFAD mice. Further reductions were observed in the case of a 12-month-old treatment group, compared with that of a 6-month-old group. Values are expressed as means ± SEM; scale bar = 100 µm and is applicable to all images. * *p* < 0.05 and ** *p* < 0.01 in comparison to WT + Vehicle and 5xFAD+TBSA.

**Figure 6 ijms-21-06664-f006:** Treatments of trans-beta-styryl-boronic acid (TBSA) compound reduced microglial activation in the cortex and hippocampus in 5xFAD mice. Six- and twelve-month 5xFAD mice and age-matched wild-type mice were treated with TBSA for 2 months, sacrificed, and their brain sections were collected and immunolabeled for IBa-1. (**A**): representative images of Iba-1-IR in the cortex, CA1 and CA3 and dentate gyrus of hippocampus from 6- and 12-month-old mice. (**C**–**E**): The number of labeled Iba-1-IR (activated microglia) was significantly reduced in the cortex (**B**,**C**), CA1 area (**D**,**E**) and CA3 area (**F**,**G**) of hippocampus in comparison to vehicle-treated 5xFAD mice. Further reductions were observed in the case of the 12-month-old mice treatment group, compared with that of the 6-month-old group. Values are expressed as means ± SEM; scale bar = 100 µm and is applicable to all images. * *p* < 0.05 and ** *p* < 0.01 in comparison to WT + Vehicle and 5xFAD + TBSA.

**Figure 7 ijms-21-06664-f007:** Effects of trans-beta-styryl-boronic acid (TBSA) compound on anxiety and activity levels in 5xFAD mice. The open-field test was used to access overall activity and anxiety in mice, both pre- and post- treatment. (**A**,**B**): The total number of fecal boli excreted during the open-field test, which is a commonly used measure of anxiety, in both the 6-month-old (**A**) and 12-month-old (**B**) group of mice increased in 5xFAD mice prior to treatment, with TBSA-treated 5xFAD excreting fewer boli than untreated 5xFAD mice during posttreatment testing. (**C**–**F**): No significant between-group differences were observed in total distance travel (**C**,**D**) and average speed (**E**,**F**) for both pre- and post- treated mice in either age group. Values are expressed as means ± SEM. * *p* < 0.05 in comparison to WT + Vehicle.

**Figure 8 ijms-21-06664-f008:** Treatments of trans-beta-styryl-boronic acid (TBSA) compound improved recognition memory in 5xFAD mice. Six- and twelve-month-old 5xFAD and age-matched control mice were tested on the novel object recognition (NOR) task after treatment with TBSA (0.5 mg/kg) for 2 months. (**A**): The 12-month-old 5xFAD mice, but not 6-month-old mice spent less time exploring the novel object than WT mice, but this recognition deficit was unaltered in 5xFAD mice treated with TBSA. (**B**): The 12-month-old 5xFAD mice showed a trend toward spending more time in exploring familiar objects than WT mice and 5xFAD + TBSA mice. The exploration index (**C**) and the discrimination index (**D**) revealed that 12-month-old 5xFAD, but not in 6-month old 5xFAD mice, that were treated with TBSA showed preservation of their recognition memory. Values are expressed as means ± SEM. * *p* < 0.05 in comparison to WT + Vehicle and 5xFAD + TBSA mice.

**Figure 9 ijms-21-06664-f009:** Treatments of trans-beta-styryl-boronic acid (TBSA) prevented spatial memory deficits in the acquisition trials of the Morris-water-maze task. After 2 months of treatment with TBSA (0.5 mg/kg) or vehicle the six- and twelve-month-old 5xFAD and age-matched control animals were tested on the Morris water maze (MWM) task for 5 days. The escape latency (**A**,**B**) was significantly longer for 6- (**A**) and 12-month-old (**B**) groups of 5xFAD mice in comparison to WT+TBSA and WT + vehicle-treated mice, with TBSA treatments preventing this deficit. The only between-group difference observed in length of swim path to find the hidden platform (**C**,**D**) was between the 6-month-old vehicle-treated 5xFAD mice and WT controls on day 5, which was prevented by TBSA treatments (**D**). No significant differences in swim speed were observed between any groups for either the 6- or 12-month-old mice (**E**,**F**). Values are expressed as means ± SEM. * *p* < 0.5 in comparison to WT + Vehicle, WT + TBSA.

**Figure 10 ijms-21-06664-f010:** Treatments of trans-beta-styryl-boronic acid (TBSA) prevented deficits in spatial memory during the probe trial in the Morris-water-maze task. After 5 days of training in the MWM, all groups of mice were given a 60-s probe trial and the latency to first entry, time spent, mean number of entries, and mean distance from the target quadrant were measured. (**A**): Graphical representation of MWM tank and different quadrants, along with location of platform. (**B**): Representative swim paths of each treatment group during the probe trial, with blue and red dots marking the starting and end points, respectively. (**C**): The mean number of entries into the target quadrant was significantly different in 5xFAD + TBSA and WT + TBSA in comparison to 5xFAD + vehicle and WT + vehicle-treated groups. (**D**): Time spent in the target quadrant was decreased for the 6-, but not the 12-month-old 5xFAD mice, and this deficit was mitigated by the TBSA treatment. (**E**): There was an increase in latency of first entry to the target quadrant for vehicle-treated 5xFAD mice, whereas TBSA treatment prevented this increase. (**F**): The 12-month old 5xFAD mice had an increase in mean distance from the platform location, which was prevented in the TBSA-treated mice. Values are expressed as means ± SEM. * *p* < 0.05 and ** *p* < 0.01 in comparison to WT + Vehicle, 5xFAD + TBSA and WT + TBSA.

**Figure 11 ijms-21-06664-f011:** Possible mechanism of anti-amyloid and antineuroinflammatory action of TBSA in animal models of AD. Schematic diagram shows the anti-amyloid and antineuroinflammatory mechanism of trans-beta-styryl-boronic acid (TBSA) which prevented neuronal damage and preserved spatial memory in 5xFAD mice.

**Figure 12 ijms-21-06664-f012:** Schematic diagram showing experimental design and treatment paradigm.

**Table 1 ijms-21-06664-t001:** Groups and number of animals for key measures.

Groups	Age Groups	Behavior	CV Stain	GFAP/Iba-1	Aβ Plaques
WT + Vehicle	6 months	12 (M = 8, F = 4)	4 (M = 2, F = 2)	3 (M = 2, F = 1)	3 (M = 2, F = 1)
5xFAD + Vehicle	6 months	12 (M = 7, F = 5)	4 (M = 3, F = 1)	6 (M = 4, F = 2)	6 (M = 4, F = 2)
5xFAD + TBSA	6 months	12 (M = 6, F = 6)	4 (M = 2, F = 2)	3 (M = 1, F = 2)	3 (M = 2, F = 1)
WT + TBSA	6 months	12 (M = 8, F = 4)	3 (M = 2, F = 1)	3 (M = 2, F = 1)	3 (M = 2, F = 1)
WT + Vehicle	12 months	11 (M = 6, F = 5)	4 (M = 2, F = 2)	3 (M = 1, F = 2)	3 (M = 1, F = 2)
5xFAD + Vehicle	12 months	10 (M = 6, F = 4)	4 (M = 2, F = 2)	4 (M = 3, F = 1)	6 (M = 4, F = 2)
5xFAD + TBSA	12 months	11 (M = 7, F = 5)	4 (M = 3, F = 1)	3 (M = 2, F = 1)	3 (M = 2, F = 1)
WT + TBSA	12 months	10 (M = 5, F = 5)	3 (M = 2, F = 1)	3 (M = 1, F = 2)	3 (M = 1, F = 2)

Note: WT + Vehicle = wild-type mice given vehicle (0.5% methylcellulose); M = male; F = female; CV = cresyl violet; GFAP = glial fibrillary acidic protein; Iba-1 = ionized calcium-binding adapter molecule 1; Aβ = amyloid beta protein; 5xFAD + Vehicle = 5xFAD mice given vehicle; 5xFAD + TBSA = 5xFAD mice given TBSA (0.5% trans-beta-styryl-boronic acid); WT + TBSA = wild-type mice given TBSA.

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
