# Peer review of "RETRACTED: Ameliorative Properties of Boronic Compounds in In Vitro and In Vivo Models of Alzheimer’s Disease"

_ijms, 2020, doi:10.3390/ijms21186664_

Round 1
Reviewer 1 Report
The manuscript titled, “Ameliorative properties of boronic compounds on in vitro and in vivo models of Alzheimer’s disease” by Maiti, P. et. al. describes the effect of two small molecules, TPVA and TBSA, on three models of Abeta aggregation and/or toxicity. Namely, they utilized both a C. elegans model and a mouse model of AD, in addition to performing in vitro experiments. The confirmation of findings in multiple animal models is commendable and a strength of this study.
Because my own expertise is with the use of C. elegans as a model system, I will focus my review on Figure 2 and allow others with more experience with mice address the other figures. In figure 2, the authors utilized C. elegans strain UA198. This strain was previously published (Treusch, Sebastian, et al. "Functional links between Aβ toxicity, endocytic trafficking, and Alzheimer’s disease risk factors in yeast." Science 334.6060 (2011): 1241-1245) but the authors do not cite the original work.
The authors of the present study examined C. elegans survival and glutamatergic tail neuron survival in Abeta-expressing animals with or without TBSA or TPVA (Fig. 2C). They never say how many such neurons wild type animals have (it’s 5) but they do show that TBSA rescues neuronal loss at day 17. However, the authors do not show us control worms without Abeta, so it is impossible to determine whether the neuronal loss observed at day 17 is caused by aging or whether it is caused by Abeta. Thus, I recommend revising this figure to include the control animals at day 17.
The authors do have the relevant control animals, as they actually show in Fig. 1A that Abeta does not affect survival at day 17 beyond that of control animals not expressing Abeta. Thus, the fact that TBSA enhanced survival only for animals expressing Abeta seems odd. Does this mean that having Abeta is actually beneficial? If the drug completely blocked the expression of Abeta, we would expect survival comprabale to “control-TBSA,” which has the lowest percent survival of all conditions. I think these data are problematic, but, at the very least, they should be discussed and not glossed over.
Methods: I am concerned about how TBSA was prepared. The authors said that they dissolved 2-5mg in 10uL. Making such tiny volumes of stock solutions is very imprecise. How to weight such tiny amounts? How to know that it all got into solution? I am even more concerned that when the authors indicate in figure 2 that they treated animals with “vehicle” they actually didn’t include any DMSO. As DMSO is the necessary solvent for the drugs in question, then DMSO must be included in mock-treatments.
I am concerned about how TBSA and TPVA were administered. The authors indicate that they spread the drugs on the top of already-solidified agar. Thus, over the weeks of an animal’s lifespan, the effective concentration will necessarily decline as the chemicals diffuse through the agar. I suggest either including the drugs in the entire volume of agar before pouring plates or performing assays in liquid culture (perhaps difficult for multi-week survival assays).
Finally, the authors poured TBSA or TPVA on top of the bacteria that serves as a food source. Do the authors know that the effect of the drugs on C. elegans survival was not due to an indirect effect on their food source?
Author Response
Reviewer-1
The manuscript titled, “Ameliorative properties of boronic compounds on in vitro and in vivo models of Alzheimer’s disease” by Maiti, P. et. al. describes the effect of two small molecules, TPVA and TBSA, on three models of Abeta aggregation and/or toxicity. Namely, they utilized both a C. elegans model and a mouse model of AD, in addition to performing in vitro experiments. The confirmation of findings in multiple animal models is commendable and a strength of this study.
Reviewer 1 Comment:
Because my own expertise is with the use of C. elegans as a model system, I will focus my review on Figure 2 and allow others with more experience with mice address the other figures. In figure 2, the authors utilized C. elegans strain UA198. This strain was previously published (Treusch, Sebastian, et al. "Functional links between Aβ toxicity, endocytic trafficking, and Alzheimer’s disease risk factors in yeast." Science 334.6060 (2011): 1241-1245) but the authors do not cite the original work.
Response: We agree and are surprised that we made this oversight! Thank you for pointing this out. This reference is now included. Please see the Ref no 15 in the revised manuscript.
Reviewer 1 Comment:
The authors of the present study examined C. elegans survival and glutamatergic tail neuron survival in Abeta-expressing animals with or without TBSA or TPVA (Fig. 2C). They never say how many such neurons wild type animals have (it’s 5) but they do show that TBSA rescues neuronal loss at day 17. However, the authors do not show us control worms without Abeta, so it is impossible to determine whether the neuronal loss observed at day 17 is caused by aging or whether it is caused by Abeta. Thus, I recommend revising this figure to include the control animals at day 17.
Response: We are happy to make reporting clearer that Peat-4: GFP animals normally have 5 GFP expressing neurons in the tail of healthy young adult to adult animals. We have now added this information to the methods and materials results (page 4, line 127), and discussion sections (page 14, line 406-407) of the revised manuscript. As the reviewer suggests, we did indeed, do all of the Peat-4::GFP control strain experiments +/- TBSA treatment and had not included these so as to streamline the figures for ease of reading, but are happy to include these data. The micrographs of the Peat-4::GFP non-Aβ background strain have now been included in Figure 2C, and the quantitative analysis of the number of glutamatergic neurons found in the tails of these control animals has been included in the bar graph Figure 2D.
Reviewer 1 Comment:
The authors do have the relevant control animals, as they actually show in Fig. 1A that Abeta does not affect survival at day 17 beyond that of control animals not expressing Abeta. Thus, the fact that TBSA enhanced survival only for animals expressing Abeta seems odd. Does this mean that having Abeta is actually beneficial? If the drug completely blocked the expression of Abeta, we would expect survival comparable to “control-TBSA,” which has the lowest percent survival of all conditions. I think these data are problematic, but, at the very least, they should be discussed and not glossed over.
Response: It is true that the survival of the Peat-4::GFP control animals treated with TBSA had the lowest survival rate at 17 days of age, but that survival was not statistically different from Peat-4::GFP untreated animals, nor the untreated C. elegans expressing Aβ42, hence, only the C. elegans expressing Aβ42 treated with TBSA had a stronger survival rate by day 17. Although the Aβ did not prove to be as detrimental as expected, it was clearly not beneficial, as the survival of untreated C. elegans expressing Aβ42 untreated was not statistically different from that of any of the other C. elegans groups, except for the Aβ42-treated group receiving the TBSA. This suggests that the TBSA drug, confers a potent support of health in the Ab expressing animals, and that in the absence of Ab, the drug may confer a low level of toxicity, without this preferred molecular target. These sentences are included in the discussion part of the revised manuscript. Please see page 14, line 411-420.
Reviewer 1 comment:
Methods: I am concerned about how TBSA was prepared. The authors said that they dissolved 2-5mg in 10 uL. Making such tiny volumes of stock solutions is very imprecise. How to weight such tiny amounts? How to know that it all got into solution? I am even more concerned that when the authors indicate in figure 2 that they treated animals with “vehicle” they actually didn’t include any DMSO. As DMSO is the necessary solvent for the drugs in question, then DMSO must be included in mock-treatments. I am concerned about how TBSA and TPVA were administered. The authors indicate that they spread the drugs on the top of already-solidified agar. Thus, over the weeks of an animal’s lifespan, the effective concentration will necessarily decline as the chemicals diffuse through the agar. I suggest either including the drugs in the entire volume of agar before pouring plates or performing assays in liquid culture (perhaps difficult for multi-week survival assays).
Response: We appreciate this attention to detail and the reviewer is clearly someone who recognizes that every aspect of treatment and consistency of methodology matters with C. elegans. We have expanded our methods section (Please see page 19, line 592-603) in order to make it clear that these issues were handled consistently and responsibly. First, the small starting quantity of drug was weighed on an analytical balance, directly into the tared tube in which the drug was initially solubilized with the 10ul drop of DMSO. To this tube, aqueous S media was added in order to attain a concentration of 1mg/ml, then carefully vortexed until all of the drug was clearly in solution. An aliquot of this 1mg/ml solution was then used to make the 10mg/ml working concentration. Fresh powdered drug from the desiccated vial was prepared this way every single time the drug was used. Again, 75mL of the drug was carefully distributed over the surface of the food and entire agar plate. Indeed, the vehicle controls were also prepared this way. Thus, in our experiments the final concentration of DMSO spread on to the plate was never greater than 0.005%. Notably, we always dosed new plates and transferred animals every 2 days so that they were continually exposed to freshly prepared drug, as well as being pulled away from their progeny so that no confusion in generations would take place. We did not drug our animals with Fluorodeoxyuridine to stop them from producing progeny as some investigators do to ease their timeline of treatments. We did not want the presence of another drug risking the interpretation of our results, and we were committed to providing fresh TBSA treatment every 2 days. As for surface placement of the drug, to cover the entire surface of the food and agar versus drug throughout the agar, our experience has indicated that in the tighter agar percent that we use (19 g/L versus 16-17 g/L), and the frequency of our new plating and dosing, drug throughout the agar is unnecessary and cost prohibitive. As the reviewer suggests, in liquid dosing, which provides a situation in which the animals cannot run, and they cannot hide, it does present its own complications in utility and interpretation of results. In addition to making it difficult to retrieve all animals for analysis, the exertion, metabolism and energy usage of constant swimming for the animals can confound ways in which the influences of pathology in aged animals is interpreted. These were confounding factors that we sought to avoid.
Response:
Reviewer 1 comment:
Finally, the authors poured TBSA or TPVA on top of the bacteria that serves as a food source. Do the authors know that the effect of the drugs on C. elegans survival was not due to an indirect effect on their food source?
Response: While we did not comment on this in the manuscript, we did observe that neither TBSA or TPVA appeared to have a positive or negative effect on the growth or consumption of the OP50 E. coli food source compared to vehicle controls. Growth of the food and clearance of the food patches was the same in each condition. If TBSA was in some way enriching the food source or toxifying the food source, we would not expect these results, and we would expect that the survival data to be different from what we see.
Reviewer 2 Report
This report by Maiti et al., described the therapeutic effects of boron-containing compounds, trans-2-phenyl-vinyl-boronic-acid-MIDA-ester (TPVA) and trans-beta-styryl-boronic-acid (TBSA) on AD animal models. TBSA prevented cognitive deficits, Abeta accumulation, neuronal morphology and glial activations. Though the background is well described and their conclusion is interesting, their analysis methods are not well described, making this report to be scientifically questionable.
Major points
- It is questionable whether dot-blot assay can really address Abeta aggregation. Their antibody, 6E10, can detect both monomer and oligomer form of Abeta. Also, concentration of drug is not described. PVDF or nitrocellulose? Alternative approach should be conducted.
- It is very strange that C. elegans model expressing Abeta42 is called AD animals. Alternative name should be used.
- While the number of mice used for histochemical and immunohistochemical experiments are very low (n=3/group), their error bar is incredibly small. The number should be increased, and shown as individual points in the graph.
- Sex difference is not fully described. All experiment data should be stratified by male and female, or adjusted for sex and female, depending on the results.
Minor
- The “5sFAD” at line 425 should be “5xFAD”.
Author Response
Reviewer-2
Comments and Suggestions for Authors
This report by Maiti et al., described the therapeutic effects of boron-containing compounds, trans-2-phenyl-vinyl-boronic-acid-MIDA-ester (TPVA) and trans-beta-styryl-boronic-acid (TBSA) on AD animal models. TBSA prevented cognitive deficits, Abeta accumulation, neuronal morphology and glial activations. Though the background is well described, and their conclusion is interesting, their analysis methods are not well described, making this report to be scientifically questionable.
Major points
It is questionable whether dot-blot assay can really address Abeta aggregation. Their antibody, 6E10, can detect both monomer and oligomer form of Abeta.
Also, concentration of drug is not described. PVDF or nitrocellulose? Alternative approach should be conducted.
Response: We agree with reviewer that 6E10 binds with monomers fibrils and we should use some other antibodies, such as oligomer specific (e.g. A11) or fibril specific antibodies (e.g. OC) to investigate anti-amyloid properties of boron compounds by dot blot assay, however, many researchers also used 6E10 to investigate overall Aβ42 aggregation pattern.
Initially we have used 5-, 10- and 20 µg of TPVA (Fig 1B-C) and 5-, 10-, 20-, 50 and 100 µg/mL of TBSA (Fig 1D-E) in separate blots. Then we compared 1, 5- and 10 µg/mL of both TPVA and TBSA in a same blot (Fig 1F-G).
We used the PVDF membrane for our dot blot assays.
To better understand the anti-amyloid properties of boron compounds we should investigate the changes of secondary structure (random coil to β-pleated sheet) of Aβ42 peptide by circular dichroism spectroscopy (CD spectroscopy), or to study the amyloid aggregation size by dynamic light scattering or transmission electron microscopy to investigate the overall morphological changes from monomer to oligomers, protofibril or fibril formation. However, due to limitations in our research facility, we could not use these techniques. As such, we cannot make strong statements about the extent of boron-induced Aβ suppression using only this dot-blot analyses we performed, but the results do lend some support for the converging evidence that the boron compounds can reduce Aβ, as shown in other assays. We have included theses informations in the “Discussion” part of the revised manuscript. Please see page no 13-14, line 374-383.
It is very strange that C. elegans model expressing Abeta42 is called AD animals. Alternative name should be used.
Response: Thanks for the suggestion. We changed this term to “C. elegans expressing Aβ42” in the revised manuscript.
While the number of mice used for histochemical and immunohistochemical experiments are very low (n=3/group), their error bar is incredibly small. The number should be increased and shown as individual points in the graph.
Response: We have mentioned the number of mice used in each parameter. Please see Table 1 in the revised manuscript. In each mouse, we have taken multiple images from each mouse to get average values which makes the error bars small.
Sex difference is not fully described. All experiment data should be stratified by male and female, or adjusted for sex and female, depending on the results.
Response: In the Table-1 of the revised manuscript, we have added the number of male and female mice used for this study.
Minor: The “5sFAD” at line 425 should be “5xFAD”.
Response: This is corrected. Please see page 16, line 473 in the revised manuscript.
Reviewer 3 Report
I enjoyed reading the research paper, Ameliorative properties of boronic compounds on in 2 vitro and in vivo models of Alzheimer’s disease. The hypothesis is well designed and experiments are well planned with perfect methodology. The data is well presented.
The paper needs the following major corrections
- Abstract needs to be modified. Introduce the disease and hypothesis before jump into result directly.
- Structure of the compounds never mentioned. Introduce structures as Fig 1.
- We suggest to do docking model to compounds to Abeta so that binding areas can be visualized.
- Fig 11, Possible mechanisms was cited. But no detailed mechanisms are explained. Binding to monomer to oligomer and fibril. How aggregation pathway inhibited. You can add all these in the mechanism from prevention of abeta formation, aggregation to behavior.
- The discussion section never explain by Boronic compounds are used and what is background work on this.
- Also add very detailed information on how Boronic compounds are unique to be drugs for AD.
Truly good study.
Author Response
Reviewer-3
Comments and Suggestions for Authors
I enjoyed reading the research paper, Ameliorative properties of boronic compounds on in 2 vitro and in vivo models of Alzheimer’s disease. The hypothesis is well designed, and experiments are well planned with perfect methodology. The data is well presented.
The paper needs the following major corrections
Abstract needs to be modified. Introduce the disease and hypothesis before jump into result directly.
Response: We have carefully gone through the abstract and modified accordingly. Please see the revised abstract.
Structure of the compounds never mentioned. Introduce structures as Fig 1.
Response: In the revised manuscript, the details structure of the boron compounds used in this study have been introduced. Please see page 2, line 71-80.
We suggest doing docking model to compounds to Abeta so that binding areas can be visualized.
Response: Previous studies have been performed and confirmed that boron compounds bind with APP and Aβ. Please see the references 4 and 34 in the revised manuscript.
Fig 11, Possible mechanisms was cited. But no detailed mechanisms are explained. Binding to monomer to oligomer and fibril. How aggregation pathway inhibited. You can add all these in the mechanism from prevention of abeta formation, aggregation to behavior.
Response: Thanks for reviewer’s suggestion. We have modified it, please see Fig 11 in the revised manuscript. Previous reports suggested that boron compounds bind with amyloid precursor protein and inhibit Aβ aggregation. Please see the references 4 and 34 in the revised manuscript.
The discussion section never explain by Boronic compounds are used and what is background work on this.
Response: We have added few sentences in the discussion part of the revised manuscript to bolster the rationale for the use of these boron compounds in AD therapy. Please see page 13, line 354-357 in the revised manuscript.
Also add very detailed information on how Boronic compounds are unique to be drugs for AD.
Response: In the discussion part of the revised manuscript, we have included new sentences to indicate why boronic compounds are unique as a potential drug therapy for AD. Please see page 13, line 335-349.
Truly good study.
Response: Thanks to the reviewer for the kind complement about our study, and thanks to all the reviewers for their insightful comments which we believe significantly improved the quality of the manuscript.
Round 2
Reviewer 1 Report
The manuscript titled, “Ameliorative properties of boronic compounds on in vitro and in vivo models of Alzheimer’s disease” by Maiti, P. et. al. describes the effect of two small molecules, TPVA and TBSA, on three models of Abeta aggregation and/or toxicity. Namely, they utilized both a C. elegans model and a mouse model of AD, in addition to performing in vitro experiments. The manuscript has been revised in accordance with peer review recommendations. As appropriate controls are now included, results discussed with more nuance, and methodology described in more detail, this reviewer is happy to recommend acceptance of the manuscript in its current form.
Author Response
Thank you for your valuable comments and suggestions to improve our manuscript.
Reviewer 2 Report
Authors did not respond with sincerity my questions for the previous version, and thus the current manuscript seems to be still scientifically unreliable.
Major points:
- Please cite solid papers from other groups using 6E10 antibody to distinguish monomer and oligomer of Abeta. If authors cannot, they should use oligomer specific antibody for dot blot assay.
- It is unclear whether they used PVDF or nitrocellulose membrane for dot blot. They described PVDF in the Result, but nitrocellulose in the Material and Method.
- Concentrations of TPVA and TBSA used in Figure 1 should be described in the Result, Figure legend or Material Methods.
- All histochemical and immunohistochemical graphs of mouse experiment (Figure 3-6) should be shown as individual points, not simple bar graph.
- What is the meaning of error bar of Kaplan-Meier Survival analysis (Fig 2A&B)?
Author Response
Please cite solid papers from other groups using 6E10 antibody to distinguish monomer and oligomer of Abeta. If authors cannot, they should use oligomer specific antibody for dot blot assay.
Response: we have added a reference where 6E10 have been used to monitor Aβ aggregation by dot blot assay. Please see reference no 41 in the revised manuscript.
It is unclear whether they used PVDF or nitrocellulose membrane for dot blot. They described PVDF in the Result, but nitrocellulose in the Material and Method.
Response: We apologize for our mistake. It should be PVDF membrane. We have corrected this in the “Material & Methods” section in the revised manuscript. Please see page 17, line 526.
Concentrations of TPVA and TBSA used in Figure 1 should be described in the Result, Figure legend or Material Methods.
Response: We have added the information of the doses of boronic compounds used for this study in the “Result” section (please see page 3, line 99-101) and in the Figure- legend (Please see page 3, line 112 and 113) in the revised manuscript.
All histochemical and immunohistochemical graphs of mouse experiment (Figure 3-6) should be shown as individual points, not simple bar graph.
Response: As per reviewer’s suggestion, all the graphs (figure 3-6) are plotted as individual points. Please see the Figure 3-6 in the revised manuscript.
What is the meaning of error bar of Kaplan-Meier Survival analysis (Fig 2A&B)?
Response: The reviewer is correct in questioning the use of error bars in the “Kaplan-Meier” analysis, as what is portrayed in the figure was a simple survival analysis plot. As such, we removed “Kaplan-Meier” reference from the figure legend -2 and indicated that the asterisk on the SEM error bar indicates that by the end of the study, significantly more mice survived in the Trans-beta-styryl-boronic acid (TBSA), group than all other groups.
Reviewer 3 Report
The paper is revised accurately. Acceptable for publication in the present form.
Author Response
Thank you for your valuable comments and suggestion to improve our manuscript.
Round 3
Reviewer 2 Report
- Regarding 6E10 antibody on dot blot assay, the cited paper used this antibody as a control (to monitor Abeta total amount) rather than to detect Abeta aggregation. Please use oligomer specific antibody like A11 in the cited paper for dot blot assay, or perform other experiments, like thioflavin assay, or normal western blotting etc. to confirm that TBSA really inhibit Abeta aggregation.
- Again and again, the concentration, such as μg/ml or μM (not the amount, like μg) of TPVA and TBSA should be described in Fig 1.
- In the revised individual plots in Figure 3-6, there are lot of issues. (1) Figure 3B and 3D are the same plots. (2) Fig 5G does not show any data of 5xFAD vehicle group. (3) in Fig 6D, though there is a significant mark between 5xFAD vehicle ground and 5xFAD TBSA group, it looks no significant difference between them. (4) Several plot, like Fig5D, Fig 6B, one group appears to have just 2 mice. If points are overlapped, please separate them horizontally.
Author Response
Regarding 6E10 antibody on dot blot assay, the cited paper used this antibody as a control (to monitor Abeta total amount) rather than to detect Abeta aggregation. Please use oligomer specific antibodies like A11 in the cited paper for dot blot assay, or perform other experiments, like thioflavin assay, or normal western blotting etc. to confirm that TBSA really inhibits Abeta aggregation.
Response: We agree with reviewer that 6E10 antibody, itself, may not be sufficient as a precise measurement of Aβ aggregation, and that an oligomer- or fibril- specific antibody would be preferable. However, in the present context we believe our dot-blot assay does add to the converging evidence that TBSA and TPVA can reduce Aβ aggregation. We have added a few sentences in the discussion of the revised manuscript along with a few more references to underscore that the dot-blot assay can provide at least some evidence, albeit not sufficient on its own, to suggest that the boron compounds can affect Aβ aggregation. Please see page no 14.
Again, and again, the concentration, such as μg/ml or μM (not the amount, like μg) of TPVA and TBSA should be described in Fig 1.
Response: We apologize for this oversight and thank the reviewer for catching it. It should be μg/ml. We have corrected these in the revised Fig 1 and in results, figure legend section (page 3) and in the discussion part (page 13) of the revised manuscript.
In the revised individual plots in Figure 3-6, there are a lot of issues. (1) Figure 3B and 3D are the same plots. (2) Fig 5G does not show any data of 5xFAD vehicle group. (3) in Fig 6D, though there is a significant mark between 5xFAD vehicle ground and 5xFAD TBSA group, it looks no significant difference between them. (4) Several plots, like Fig5D, Fig 6B, one group appears to have just 2 mice. If points are overlapped, please separate them horizontally.
Response: Thanks, the reviewer for catching those mistakes
- Fig 3B and 3D are corrected in the revised manuscript
- We have corrected the Fig 5G in the revised manuscript
- We have corrected the Fig 6D in the revised manuscript
- The graphs of these plots are made with three mice and some of the data were overlapped. (please see the number of mice used for analyses of the graph in Table 1.